# Ecological Connectivity in Agricultural Green Infrastructure: Suggested Criteria for Fine Scale Assessment and Planning

**Simone Valeri \*, Laura Zavattero and Giulia Capotorti**

Department of Environmental Biology, Sapienza University of Rome, P.le Aldo Moro 5, 00185 Rome, Italy; laura.zavattero@uniroma1.it (L.Z.); giulia.capotorti@uniroma1.it (G.C.)
\* Correspondence: simone.valeri@uniroma1.it

**Abstract:** In promoting biodiversity conservation and ecosystem service capacity, landscape connectivity is considered a critical feature to counteract the negative effects of fragmentation. Under a Green Infrastructure (GI) perspective, this is especially true in rural and peri-urban areas where a high degree of connectivity may be associated with the enhancement of agriculture multifunctionality and sustainability. With respect to GI planning and connectivity assessment, the role of dispersal traits of tree species is gaining increasing attention. However, little evidence is available on how to select plant species to be primarily favored, as well as on the role of landscape heterogeneity and habitat quality in driving the dispersal success. The present work is aimed at suggesting a methodological approach for addressing these knowledge gaps, at fine scales and for peri-urban agricultural landscapes, by means of a case study in the Metropolitan City of Rome. The study area was stratified into Environmental Units, each supporting a unique type of Potential Natural Vegetation (PNV), and a multi-step procedure was designed for setting priorities aimed at enhancing connectivity. First, GI components were defined based on the selection of the target species to be supported, on a fine scale land cover mapping and on the assessment of land cover type naturalness. Second, the study area was characterized by a Morphological Spatial Pattern Analysis (MSPA) and connectivity was assessed by Number of Components (NC) and functional connectivity metrics. Third, conservation and restoration measures have been prioritized and statistically validated. Notwithstanding the recognized limits, the approach proved to be functional in the considered context and at the adopted level of detail. Therefore, it could give useful methodological hints for the requalification of transitional urban–rural areas and for the achievement of related sustainable development goals in metropolitan regions.

**Keywords:** Peri-urban landscapes; Metropolitan areas; MSPA; fragmentation; native woody species; environmental units; naturalness; ecological corridors; conservation and restoration priorities.

## 1. Introduction

Connectivity represents an emergent property of landscapes with respect to species dispersal and ecological processes [1,2]. As such, it is increasingly recognized as a fundamental feature for enhancing biodiversity conservation and ecosystem service capacity against fragmentation, in both ecological networks and GI planning [3,4]. These roles of connectivity have been quite thoroughly disentangled in urban areas as well as in rural landscapes [5–7], while additional values are emerging for peri-urban transitional contexts, spanning from the reconnection between cities and their countryside to the enhancement of agriculture multifunctionality and sustainable development of metropolitan regions [8,9]. Pragmatically, ecological connectivity analyses focus on structural, functional, and dynamic individual characteristics and mutual relationships between patches, matrix, and corridors in order to assess landscape permeability to species movement [5,10].

As regards agricultural landscapes, current research is increasingly addressing the vegetation component of biodiversity in addition to the faunistic one, which represents a more traditional target of investigation [11]. Both the impact of plant community composition on connectivity [12] and, vice versa, the impact of connectivity features on taxonomic and functional structure of plant communities [13] have been explored. Native status and dispersal traits of plants, corridor suitability, and patch/matrix resistance to dispersal represent the more frequently investigated attributes at the species, community, and landscape level [14–16]. Nevertheless, especially in the context of the European GI Strategy implementation [17], little evidence is available on how to select plant species to be primarily favored in dispersal and on the role of environmental heterogeneity and quality of habitat patches and corridors in facilitating/impairing such a dispersal.

The present work is aimed at suggesting a methodological approach for addressing these knowledge gaps at fine scales and for peri-urban agricultural landscapes. The approach was tested in a Natural Reserve in the Metropolitan City of Rome (Italy), within which urbanization pressure and rural landscape homogenization may impair the resilience of the rural system and its capacity to provide valuable ES despite the legally protected status [18,19]. Our findings suggest that, in such a context, the prioritization of GI actions for enhancing biodiversity and connectivity may be suitably driven by i) the selection of target plant species according to the vegetation potential, ii) the stratification of land into homogeneous environmental units, and iii) the assessment of naturalness of the landscape mosaic components.

## 2. Materials and Methods

### 2.1. Study Area

The Marcigliana Nature Reserve is located in the northeastern peri-urban sector of the Metropolitan City of Rome (42°00′18.72″N 12°35′13.92″E / 42.0052°N 12.5872°E), Italy, and covers an area of 4,696 hectares (Figure 1). It belongs to a system of protected areas in the Municipality of Rome, managed by the RomaNatura regional body, that hosts biodiversity of conservation interest at the species, ecosystem and/or genetic level (L.R. n. 29/97). The Reserve, as the whole municipality, is embedded within the ecoregional subsection of the "Roman Area", characterized by coastal Mediterranean and hilly transitional bioclimate, composite sedimentary and volcanic litho-morphology, and prevailing PNV for deciduous oak forests [20].

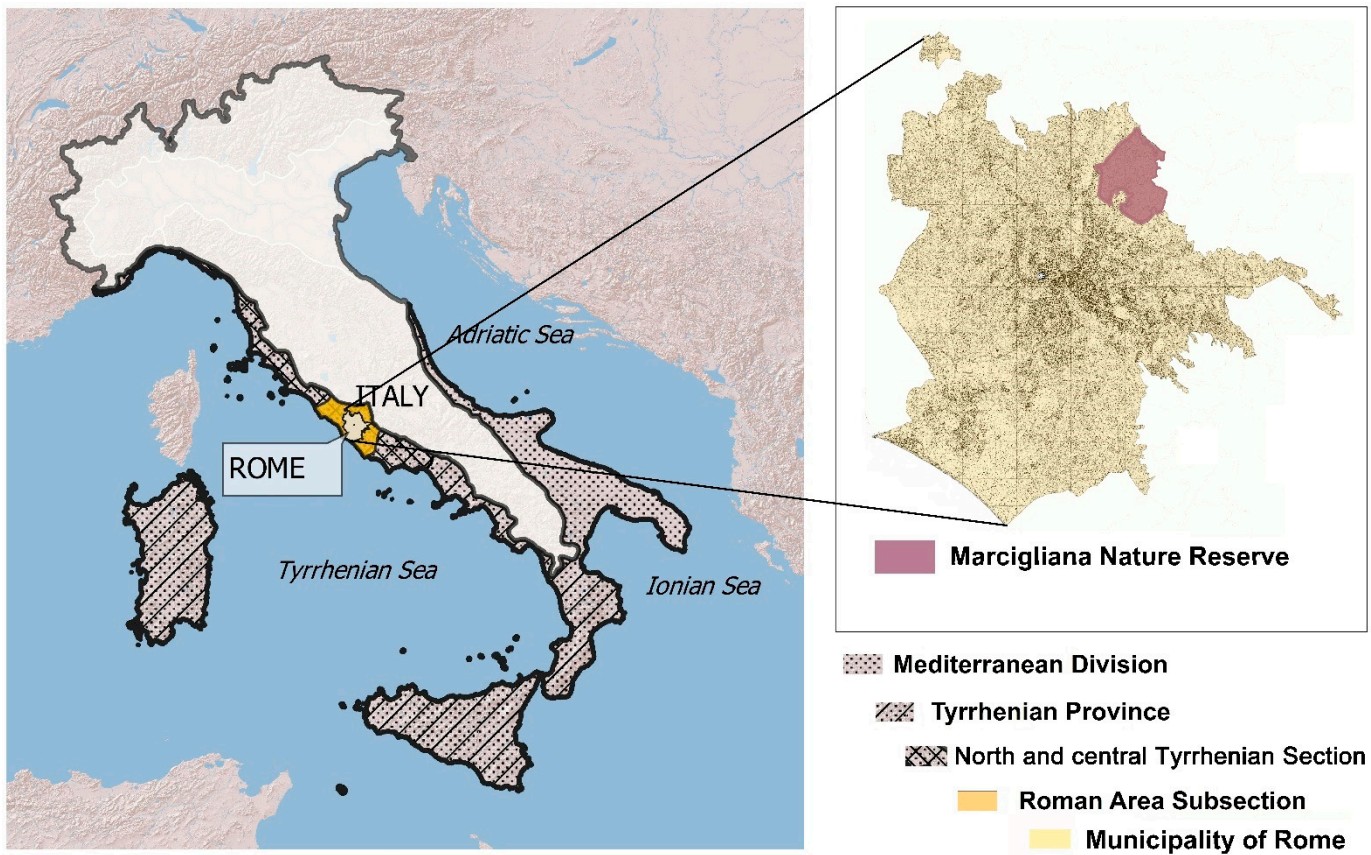

**Figure 1.** Study area. Ecoregional setting of the Municipality of Rome, from the division to the subsection level (on the left), and location of the Marcigliana Nature Reserve within the Municipality of Rome (on the right).

More in detail, the Reserve shows a varied pattern of different Environmental Unit types (EUN), i.e., homogeneous portions of land, with respect to climatic, lithologic, morphologic and PNV features, hosting a unique type of mature vegetation together with semi-natural and anthropogenic seral stages. The occurring EUNs include: i) Volcanic Plateaux (VPL), supporting Turkey oak and eastern hornbeam forest potential (*Carpino orientalis-Quercetum cerris* vegetation series) (66% of the site); ii) Alluvial Valleys (AV), supporting hygrophilous and meso-hygrophilous forest potential (*Querco roboris-Ulmetum minoris / Salicetum albae* vegetation complex) (17%); and iii) Sandy-Clayey Slopes (SCS), supporting Virgilian oak and Turkey oak forest potential (*Carpino orientalis-Quercetum cerris varietas quercetosum virgilianae* vegetation series) (17%) [21]. With respect to this potential arrangement, the present land use and land cover is starkly dominated by agricultural areas, without clear trends upon abandonment [22]. On the contrary, urban sprawl and soil consumption are threatening the rurality of the Reserve especially at its borders [23,24], with artificial areas representing about 4% of the site. Natural and semi-natural vegetation is therefore reduced to minor remnants, with the mature stages of the most widespread vegetation series types, i.e., *Quercus cerris* woods, accounting for about 10% of the site. Owing to the agricultural vocation, environmental protection rules, recognized role as a metropolitan ecological network buffer zone, and geographic position between the consolidating city and traditional rural landscapes of the countryside [25–27], the Reserve has been selected as a suitable case study for addressing the connectivity issue in support of peri-urban GI planning.

*2.2. Research Design*

In keeping with the principles proposed for local scale GI planning [28], a multi-step procedure was designed for setting priority measures aimed at enhancing the ecological connectivity in a peri-urban agricultural landscape (Figure 2).

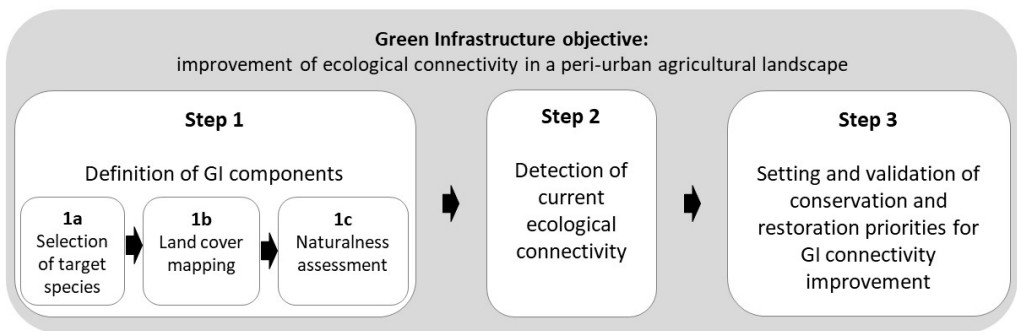

**Figure 2.** Multi-step procedure aimed at setting conservation and restoration priorities for GI connectivity improvement in the study area.

First, the current GI components were defined based on the selection of the target species to be supported (step 1a), on a fine scale land cover mapping (including natural and semi-natural ecosystem patches as well as linear vegetation elements) (step 1b), and on the assessment of land cover type naturalness (step 1c). Second, current ecological connectivity was assessed in both structural and functional terms (step 2) and, third, conservation and restoration measures have been prioritized and validated by means of statistical correlation with the observed occurrence of target species (step 3).

More in-depth information on the definitions of ecosystem naturalness and ecological connectivity adopted for the research [29–33] is provided in Table S1 of Supplementary Material.

### 2.3. First Step: Definition of GI Components According to Target Species, Ecosystem Occurrence, and Naturalness

Assuming that the dispersal of trees representative of the mature vegetation communities may facilitate the resistance and resilience of natural forest ecosystems in a rural landscape [34], the woody plants with a limited dispersal capacity and that are characteristic of the PNV types occurring in the Marcigliana Natural Reserve have been selected as target species (step 1a, Figure 2). These include three oak species, namely *Quercus cerris*, *Q. robur*, and *Q. virgiliana*, that are barochore and zoochore and may be effectively dispersed by the jay (*Garrulus glandarius* L.) or by hoarding rodents [35]. Since the presence of the jay in the Reserve is not ascertained [36], it was assumed that occurring small rodents, such as *Apodemus sylvaticus* L., may act as main dispersers [37,38]. The dispersal distance mediated by the wild mouse increases, up to a little more than 100 m, as the number of successive movements increases (re-dispersal) and is more affected by the distance from shelter habitats rather than by the weight of the acorn [39].

Current GI components have been then recognized according to the capacity of different land cover types to sustain the persistence, dispersal or spontaneous colonization of target species. Therefore, all the ecosystems occurring in the study area have been mapped in a GIS environment (Quantum GIS) (step 1b, Figure 2) and typified. For a finer scale definition of the GI components, ecosystem and other land cover typology was defined by detailing the legend classes of the Actual Vegetation Map of the Province of Rome (1:25,000 scale) [40]. Based on these detailed classes, an original map was drawn at 1:2,000 scale by means of Google Satellite Imagery visual interpretation, with a minimum mapping unit of 0.15 ha. The woody hedgerows occurring in the agricultural matrix, important for target species as natural and semi-natural ecosystem patches, were first drawn as polylines, then converted into polygons by a 5 m buffer either side and finally integrated in

the main map. Photointerpretation was validated with field checks for all the accessible sites, and with open-source geo-visualization tools (Google Street View and Bing Maps) and comparison with the Forest Copernicus High Resolution Layer [41] for inaccessible sites.

Both the areal and linear elements occurring in the landscape mosaic and dominated by woody species have been assumed as suitable habitats for oak persistence and dispersal, but their performance was supposed to be conditioned by the respective degree of naturalness. Specifically, naturalness has been assessed accounting for the physiognomic and structural features of the mapped woody elements with respect to those of the PNV [42] (step 1c, Figure 2; Table S1 of Supplementary Material): areal and linear elements dominated by non-native species and/or with a regular structure due to plantation activities were considered less natural than those dominated by the native species typical of the PNV and showing a spontaneous cover pattern.

*2.4. Second Step: Detection of Current Connectivity*

Current structural and functional connectivity was investigated at different levels of detail by considering as suitable habitats either just areal or both areal and linear components, and whether or not their degree of naturalness is accounted for:

- Level 1—Areal components, with both high and low degree of naturalness;
- Level 2—Both areal and linear components, with all degrees of naturalness;
- Level 3—Areal components with just a high degree of naturalness;
- Level 4—Both areal and linear components with just a high degree of naturalness.

Moreover, the three EUNs occurring in the Reserve (i.e.: VPL, with *Quercus cerris* and *Carpinus orientalis* forest potential; AV, with meso- and hygrophilous forest potential; and SCS, with *Quercus virgiliana* and *Q. cerris* forest potential) were individually investigated at the level assumed as most suitable among these four (Level 4). Thus, the 7 maps (one for each level of investigation, and three for the Level 4 stratified per EUN) were converted into binary rasters (1 = habitat; 0 = non habitat) with a spatial resolution of 5 m.

For structural connectivity detection, a MSPA along with a Network Analysis were performed. MSPA, a useful tool for describing pattern structures and automatically detecting connectivity pathways, was carried out by means of the GUIDOS Toolbox [43] with the following settings: 8-connectivity, so that foreground connectivity was based on both border and corner sharing between pixels of habitats (that sometimes have a very small extent in the source map); Transition turned on, so that more importance was posed on the role of linear elements as connectors rather than on the continuity of patch edges; Intext = 1, so that the perforations of habitat patches due to enclosed features, very rare in the study area, were neglected; Edge width = 2 pixels (10 m), so that linear elements were prevented to be recruited as areal habitats. The MSPA returned a categorization of the habitats into cores, islets, perforations, edges, loops, bridges and branches. With the same GUIDOS Toolbox, a Network Analysis was performed in order to estimate the NC in the landscape mosaic. An individual component represents a region of interconnected nodes and links, respectively generated by core and bridge MSPA categories, so that a landscape can be considered as more connected as the NC is fewer [44,45].

For functional connectivity assessment, the Integral Index of Connectivity (IIC) was estimated (Conefor 2.6 software). The index, widely recommended for habitat and link prioritization [46], provides a measure of connectivity between nodes according to a threshold distance. Given the wild mouse-mediated dispersal capacity of the target species, such a distance was approximated at 100 m. The IIC varies between 0 and 1 and positively increases with connectivity:

$$IIC = \frac{\sum_{i=1}^{n} \cdot \sum_{j=1}^{n} \cdot \dfrac{a_i a_j}{1 + n l_{ij}}}{A_L^2} \tag{1}$$

where $n$ is the total number of nodes in the landscape, $a_i$ and $a_j$ are the attributes (i.e., the extent) of nodes $i$ and $j$, $nl_{ij}$ is the number of links in the shortest path (topological distance) between patches $i$ and $j$, and $A_L$ is the maximum landscape attribute (i.e., the extent of a habitat patch covering all the landscape).

*2.5. Third Step: Prioritization of Conservation and Restoration Measures*

By combining multiple indicators, alternatively fitting with areal or linear components, conservation priorities for the maintenance of landscape connectivity were assigned to habitat patches and corridors at the Level 4 stratified per EUN. The values for each indicator were then scored and added together for the assignment of a comprehensive priority to each component.

Specifically, habitat patches were prioritized according to:

a)  Node Importance [47], calculated as

$$IIC\,(\%) = 100 \, \cdot \frac{IIC - IICremove}{IIC} \tag{2}$$

where *IIC* is the index value when the overall existing nodes are considered, and *IICremove* is the index value after the removal of that single node from the landscape. Priority scores for Node Importance were assigned following the distribution of the indicator values into quartiles;

b)  Condition of the EUN of occurrence, derived from the previous methodological step and qualitatively scored, with a null value assigned to the less critical EUNs and a unit value assigned to the most critical one.

Corridors, prevalently links (*bridges*), but also the other linear MSPA categories, were evaluated by means of:

b)  Condition of the EUN of occurrence, as for nodes;

c)  Link Removal indicator, so that the removal of each *bridge* was simulated, the respective impact (dIIC) calculated as for nodes, and the priority quantitatively scored according to distribution of the indicator values into quartiles;

d)  Conservation priority of the nodes connected by the link, derived from Node Importance (criterion a) and qualitatively scored in compliance with every emerging combination (i.e., the higher the importance of nodes, the higher the score assigned to the connector);

e)  Connection importance, assigned to links that, if removed, originate a new component;

f)  Structural contiguity and singularity of connections, so that a higher priority was assigned to bridges and branches with respect to isolated islets (due to less contiguity) and to loops (due to connection redundancy).

In order to validate conservation priorities for links, the presence and abundance of *Quercus* specimens were estimated by means of physiognomic structural surveys of the linear woody elements at accessible sites. Subsequently, the correlation between abundance and conservation priority was assessed with the Kendall Tau-b statistic [48], whose values range from −1 (100% negative association) to +1 (100% positive association) with 0 indicating absence of association. The Kendall Tau-b coefficient is defined as:

$$\tau_B = \frac{f_c - f_d}{\sqrt{(f_c + f_1 + E_x)(f_c + f_1 + E_y)}} \tag{3}$$

where $f_c$ are the concordance frequencies; $f_d$ is the frequency of discrepancies; $E_x(y)$ are the bonds of the independent (and dependent) variable. Owing to the difficulties encountered in making many surveys, it was reasonable to set a level of significance $p \leq 0.10$.

With respect to restoration, the criteria for setting priorities were defined in order to minimize conflicts with primary production [49,50], so that the boosting of links, especially the conversion of branches into bridges, was preferred to the creation of new forest patches. Moreover, such a conversion was simulated by favoring restoration of tree cover in pre-existing paths or along linear element residuals between cultivated fields (such as unpaved road edges or grass verges) (Figure 3) and by limiting the development of redundant links between nodes (i.e., loops).

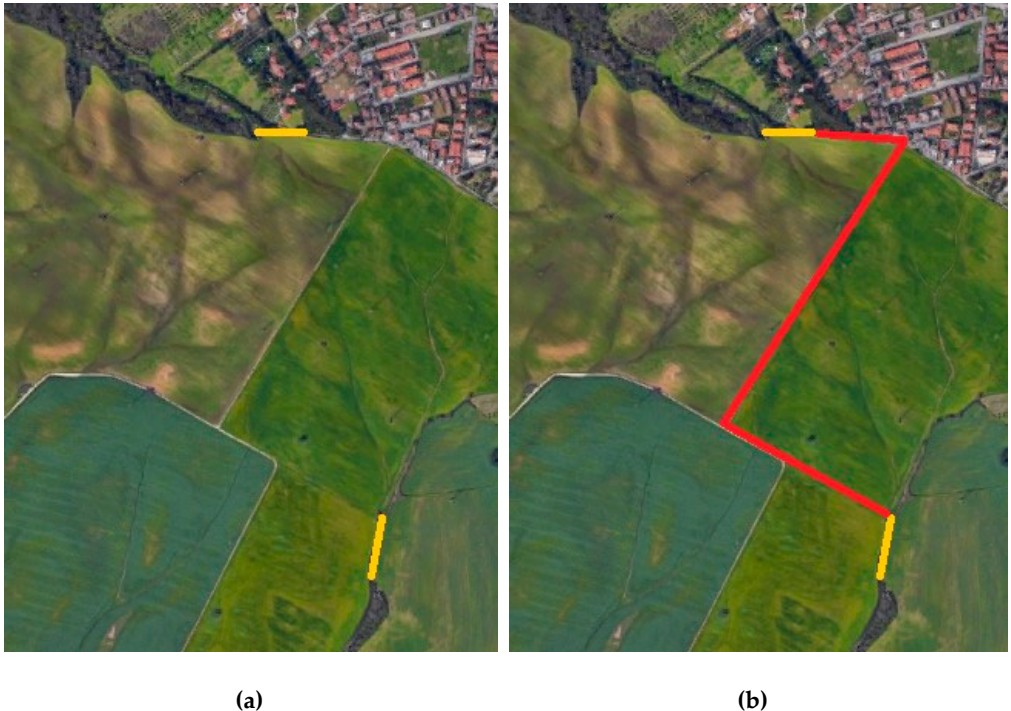

|  |  |
|:---:|:---:|
| **(a)** | **(b)** |

**Figure 3.** Simulated conversion of two *branches* (a) into a single *bridge* according to a pre-existing path (b).

The improvement in connectivity, potentially determined by the simulated restoration, was then assessed by means of Conefor connector-based (not distance-based) measures. Namely, the IIC and the NC were measured ex ante and ex post the conversion of branches into bridges.

## 3. Results

### 3.1. Current GI components

The landscape matrix of the Reserve is represented by agricultural surfaces (80%, mainly arable lands), with interspersed natural patches (16%, mainly *Quercus cerris* woods), artificial surfaces (4%, prevalently with constructions related to agricultural activities) and woody linear elements (206, 163 of which are natural and 43 artificial, with a density of 14,78 m/ha) (Figure 4).

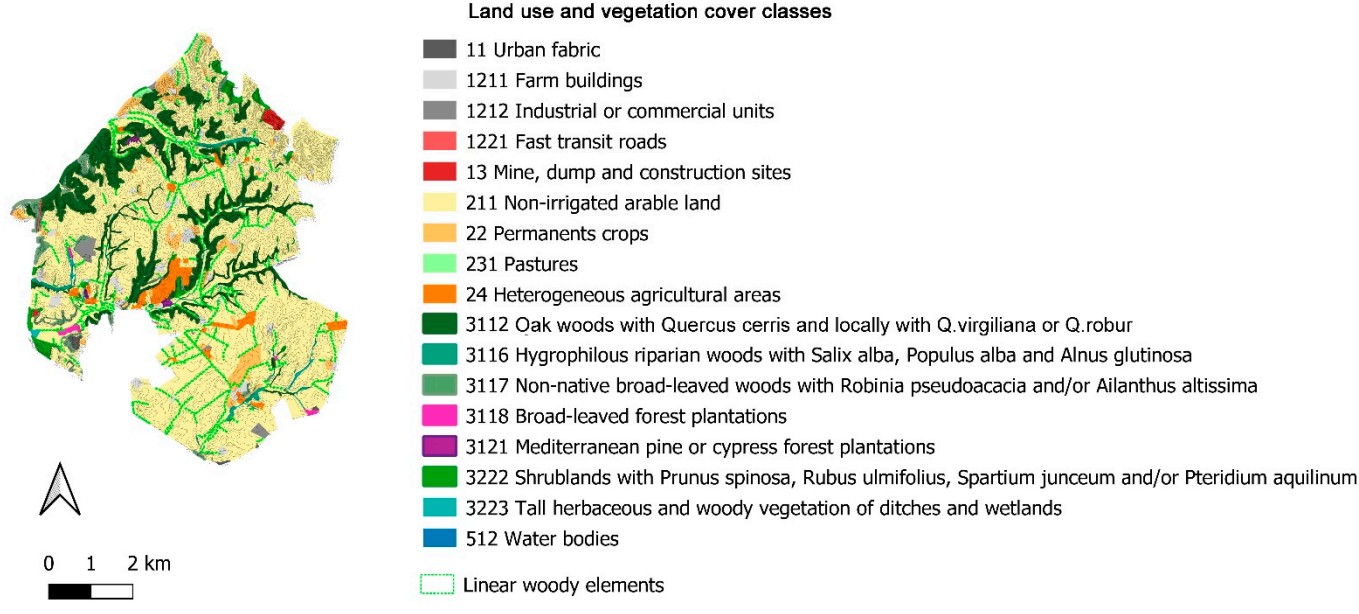

**Figure 4.** Land use and vegetation cover in the Marcigliana Nature Reserve.

Such an arrangement is coarsely confirmed at the EUN level, but with a varying prevalence of agricultural surfaces over natural vegetation and a different density of linear elements (Table 1).

**Table 1.** Landscape features (percent coverage of main land cover types and density of linear woody elements) of the EUNs occurring in the Marcigliana Nature Reserve.

| Environmental Unit | Total surface (ha) | Agricultural surfaces (%) | Natural surfaces (%) | Artificial surfaces (%) | Density of linear elements (m/ha) |
|---|---|---|---|---|---|
| Volcanic Plateaux (VPL) | 2719 | 86% | 10% | 4% | 11,7 |
| Alluvial Valleys (AV) | 602 | 75% | 22% | 3% | 44,5 |
| Sandy-Clayey Slopes (SCS) | 1138 | 65% | 32% | 3% | 6,5 |

In all the EUNs, the woody vegetation types, together with the linear woody elements occurring in the agricultural matrix, have been a priori selected as suitable GI components for supporting native oak species. According to their naturalness, these components were arranged into the following classes (Figure 5):

- areal "Natural-high", including Oak woods with *Quercus cerris* and locally with *Q. virgiliana* or *Q robur* (map code 3112); Hygrophilous woods with *Populus* sp.pl., *Salix* sp.pl., *Alnus glutinosa* and *Fraxinus oxycarpa* (3116); Shrublands with *Prunus spinosa, Rubus ulmifolius, Spartium junceum* and/or *Pteridium aquilinum* (3222); and Tall herbaceous and woody vegetation of ditches and wetlands (3223);
- areal "Natural-low", including Non-native broad-leaved woods with *Robinia pseudoacacia* and/or *Ailanthus altissima* (3117); Broad-leaved forest plantations 3118); and Mediterranean pine or cypress forest plantations (3121);
- linear "Natural", when dominated by spontaneous woody species;
- linear "Artificial", when dominated by planted woody species.

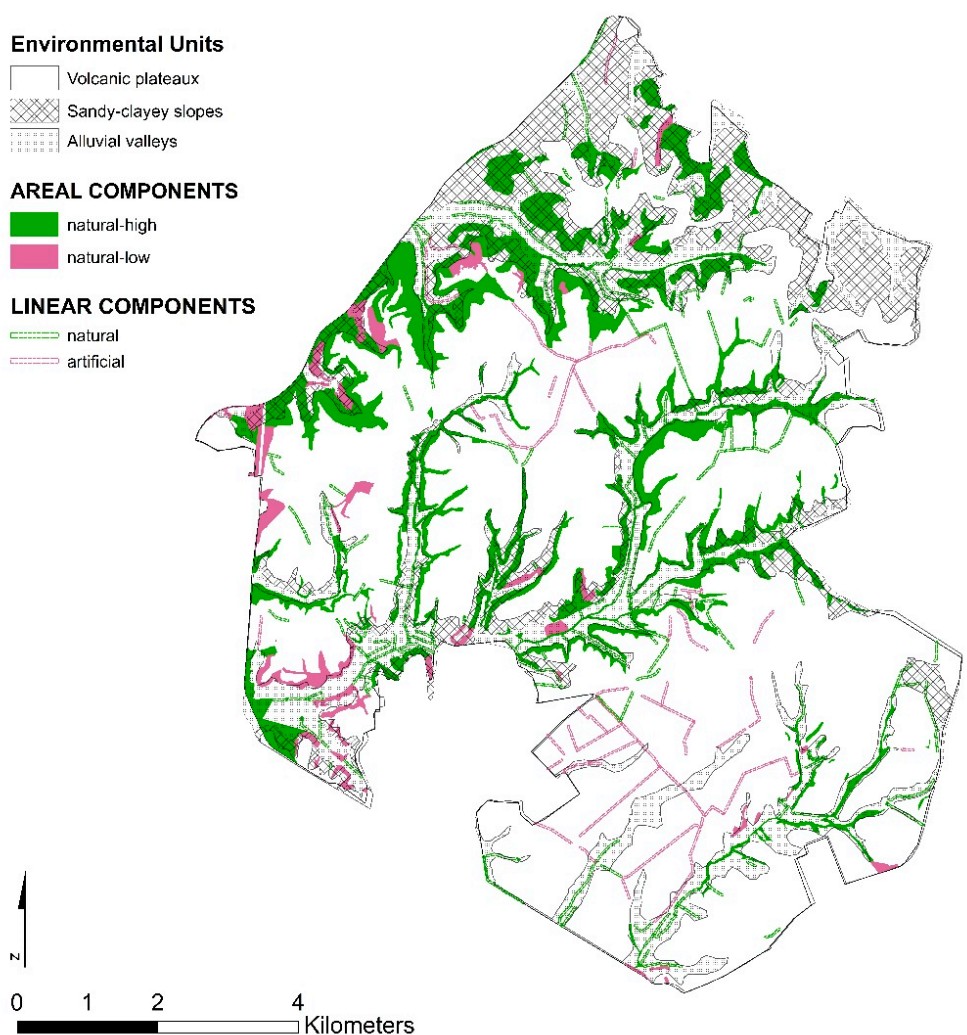

**Figure 5.** Spatial arrangement of suitable GI components in the environmental units of the Marcigliana Nature Reserve, distinguished according to structural features (areal or linear extent) and degree of naturalness.

### 3.2. Current Ecological Connectivity

Both structural connectivity features, in terms of absolute frequency of MSPA classes and NC, and functional connectivity features, in terms of IIC, for each of the four levels of investigation and for the three different EUNs at the Level 4, are reported in Table 2.

**Table 2.** Structural and functional connectivity features for the alternative levels of investigation. Note that, due to the "edge" parameter, many of the narrow forest ecosystems occurring along narrow slopes and valleys were eroded and fragmented. Therefore, the number of MSPA cores, branches, and bridges far exceeds the number of patches and linear elements in the original map.

| Connectivity fea-ture. | Level 1 (all areal components) | Level 2 (all areal and linear components) | Level 3 (natural-high areal components) | Level 4 (natural-high areal and linear components) | Level 4 / VPL (Volcanic Plateaux) | Level 4 / AV (Alluvial Valleys) | Level 4 / SCS (Sandy-Clayey Slopes) |
|---|---|---|---|---|---|---|---|
| Number of MSPA CORES | 300 | 332 | 265 | 281 | 194 | 275 | 136 |

| Number of MSPA ISLETS | 7 | 64 | 7 | 57 | 205 | 92 | 72 |
|---|---|---|---|---|---|---|---|
| Number of MSPA EDGES | 179 | 200 | 146 | 168 | 192 | 239 | 180 |
| Number of MSPA LOOPS | 0 | 12 | 1 | 8 | 19 | 30 | 22 |
| Number of MSPA BRIDGES | 119 | 194 | 105 | 161 | 56 | 141 | 42 |
| Number of MSPA BRANCHES | 1132 | 1506 | 969 | 1279 | 535 | 734 | 310 |
| Number of Components (NC) | 78 | 55 | 67 | 50 | 107 | 77 | 82 |
| Integral Index of Connectivity (IIC) | 0.004 | 0.017 | 0.002 | 0.008 | 0.00083 | 0.00503 | 0.00504 |

In the entire Reserve, and independently from the level of investigation, a conspicuous number of cores and NC is observed (with respect to $NC_{min} = 1$), denoting a high degree of habitat fragmentation. Moreover, the branches are always much more numerous than bridges, showing a high degree of discontinuity in existing corridors and further contributing to the observed NC. With respect to these general features, when linear elements are definitely taken into account (Levels 2 and 4 *vs* Levels 1 and 3), the increase in connectivity is denoted by: i) a higher number of continuous and discontinuous corridors (i.e., bridges and branches) and a consequent fewer NC; ii) a higher number of cores, showing the potential role of linear elements as habitat providers themselves, even with a high degree of naturalness (Level 4); and iii) a quadrupled values of the IIC. Alternatively, when the degree of naturalness of habitat patches and corridors is explicitly considered (Levels 3 and 4 *vs* Levels 1 and 2), the effect of quality can be distinguished from that of quantity. In this case, the decrease in NC does not indicate a better structural connectivity, but rather the complete attrition of useful components for the dispersal of target species, also denoted by the halving of the IIC.

Finally, the comparison between the three different EUNs allowed ecological connectivity features to be spatially contextualized, and VPL to be recognized as the most critical EUN with respect to AV and SCS. Actually, in VPL: i) the number of cores is not the highest but the islets are much more numerous, denoting a higher level of fragmentation and shrinkage in habitat patch dimension; ii) the ratio between cores and bridges is higher (3.46 with respect to 1.95 in AV and 3.24 in SCS) as well as the NC, denoting a more marked isolation between residual habitats; and iii) the IIC is six times lower than that of the other two EUNs, highlighting a low degree of connectivity also in functional terms.

### 3.3. Conservation Priorities

The ranking of adopted indicators, for the assignment of conservation priority scores to areal and linear GI components, is summarized in Table 3. The comprehensive conservation priority of each GI component, derived from the sum of partial indicator scores and ranked in 5 classes from 'very low' to 'very high', is instead represented in Figure 6.

**Table 3.** Conservation priority scores assigned to GI components, at the level 4 stratified per EUN, according to individual indicators (a and b for areal components, and from b to f for linear components, respectively).

| Conservation priority score | a) Node importance | b) EUN condition | c) Link removal | d) Priority of the connected nodes | e) Importance of connection | f) MSPA class |
|---|---|---|---|---|---|---|
| 5 | dIIC>11.57 (upper outliers) | | | two 'very high' priority nodes | | |
| 4 | 8.74<dIIC<11.56 (4th quartile) | | | at least one 'very high' priority node; two 'high' priority nodes | | |
| 3 | 5.05<dIIC<8.73 (3rd quartile) | | dIIC>13.59 (upper outliers) | at least one 'high' priority node; two 'medium' priority nodes | | Bridge |
| 2 | 2.30<dIIC<5.04 (2nd quartile) | | 1.30<dIIC<4.84 (from 1st to 4th quartile) | at least one 'medium' priority node | | Branch |
| 1 | 1.53<dIIC<2.29 (1st quartile) | VPL | dIIC<1.00 | two 'low' or 'very low' priority nodes | the link removal splits a component | Islet and Loop |
| Null (0) | dIIC<1.00 | SCS; AV | | | the link removal does not split any component | |

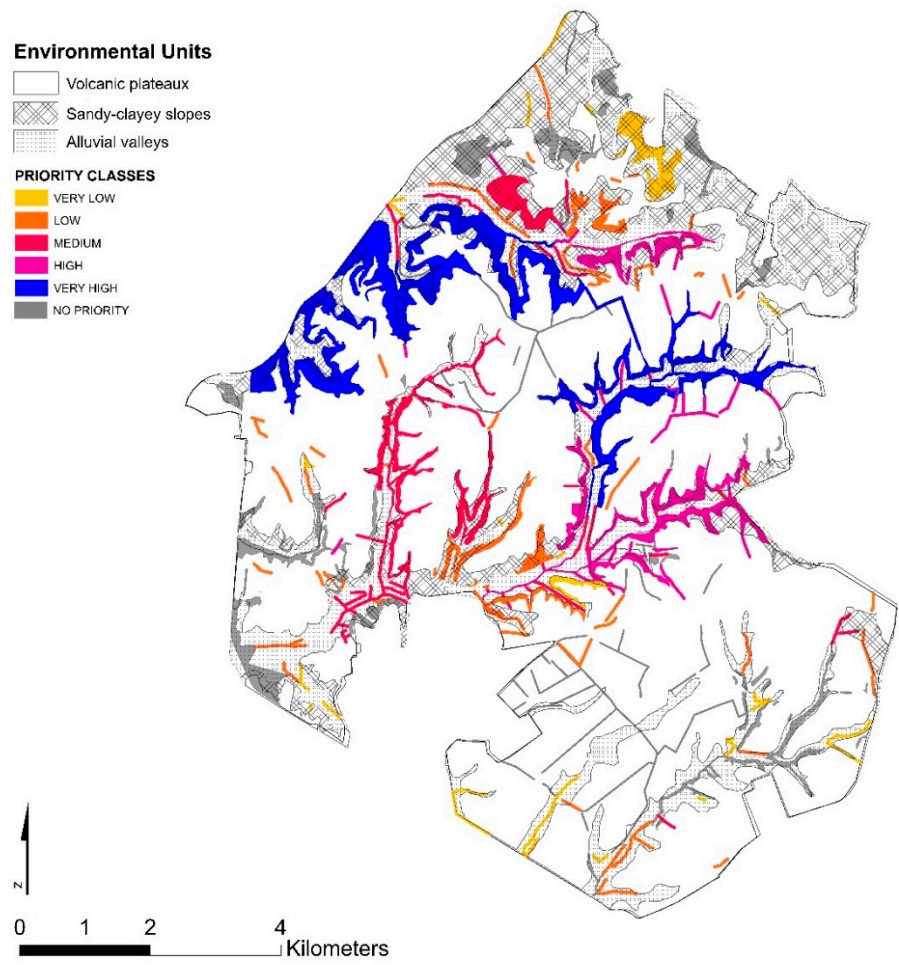

**Figure 6.** Distribution of comprehensive conservation priority of GI components in the Marcigliana Nature Reserve.

Areal elements with a positive priority are 27. Notwithstanding those with maximum values are the largest ones, some medium-size (between 10 and 25 ha) and small-size patches (<10 ha) could be prioritized as well. Linear elements with a positive priority are 123 out of the 164 natural ones. For these GI components, 40 physiognomic-structural surveys were carried out. This surveys returned a prevalence of *Rubus ulmifolius* and *Prunus spinosa* shrub formations with oak specimens occurrence in 63% of the cases (25 linear elements with *Quercus cerris*, *Q. virgiliana*, and/or *Q. robur*). The Kendall Tau-b correlation showed a significant relationship (p-value = 0.052) between the abundance of the target species in linear elements and their conservation priority.

### 3.4. Restoration Priorities

By avoiding the encroachment on existing cultivated fields and the creation of connectivity loops, the conversion of *branches* into 20 new bridges was simulated. Notwithstanding the exiguous number of simulated new links, the conversion would lead to an ecological connectivity improvement of 79% in terms of IIC (from 0.008 to 0.014) and of 14% in terms of NC (from 50 to 43).

## 4. Discussion

A methodological approach was developed and tested for addressing the improvement of GI connectivity in peri-urban agricultural landscapes. The approach was first based on fine-scale environmental stratification into homogeneous EUNs, each

supporting a unique type of PNV. Second, it was based on an in-depth definition of GI components, including linear woody elements, and, third, on the assessment of their naturalness.

With respect to approaches based on less detailed information [51], the greater mapping and assessment effort allowed some critical issues to be faced, especially pertaining to i) a focused selection of target plant species to be favoured by GI connectivity, ii) the reliability of structural and functional connectivity estimates, and iii) the steering of conservation and restoration measure prioritization.

### 4.1. Strength and Weakness of Target Species Selection

The selection of target plant species was based upon the recognition of PNV types, so that not only limited dispersal ability but also representativeness of the varied ecological potential of the site has been considered. In the study area, different species of the genus *Quercus* comply with both these requirements. Their conservation and facilitation can thus actively contribute to boost native biodiversity, control biological invasions, facilitate ecological and biogeographic coherence of landscape management measures, and guarantee a high level of restoration success in a peri-urban rural landscape [52–54]. Moreover, even though these aspects have not been deepened and go beyond the objectives of the work, oak species are expected to play a crucial role for rural landscape resilience and agriculture sustainability as keystone components of mature vegetation communities [55,56]. Therefore, they should preferentially contribute to achieve GI multifunctionality with respect to species selected for their endemic, rare or threatened status and usually targeted for the exclusive objective of biodiversity conservation in ecological network design [57].

However, some factors may limit the restoration success for these target species, such as livestock overgrazing, intensive pruning, and shrub clearance [58–61]. These constraints should be carefully considered, especially in a prospective implementation phase, and eventually mitigated by coupling oak plantation and seeding with shrub restoration.

### 4.2. Strength and Weakness of Connectivity Assessment

As regards connectivity, the estimates performed at different levels of detail confirmed the significance of explicitly accounting for the occurrence of linear landscape elements in rural contexts, as matrix permeability enhancers [62–64]. Similarly, differences in estimates due to the varied naturalness of landscape mosaic components have been documented, complementing the evidence recently arising from broader scale investigations [7,65,66].

The limited number of alternative observations prevented however to test the statistical significance of such differences, so that more alternative settings and/or a comparison with similar case studies have to be explored for deepening knowledge in this respect. Moreover, the historical persistence of occurring hedgerows could be analyzed for strengthening the assessment of corridor effectiveness [67].

### 4.3. Strength and Weakness of Prioritization Procedure

For prioritization, all the collected information on environmental stratification and habitat and landscape condition was capitalized by means of an additive assessment, as already experimented but with different criteria and for different landscape contexts [68,69]. Accordingly, conservation and restoration measures were not only defined on the basis of connectivity metrics, but also differentiated accounting for the conservation status of the EUNs (i.e., the varying fragmentation degree due to differences in environmental suitability for intensive land uses), the naturalness of the occurring elements (avoiding a GI design just based on structural land cover information), and the current availability of ecological corridors for target species (in both structural and functional terms). Other

authors already highlighted that ecological connectivity varies according to landscape types, without however incorporating such an information in the prioritization process [70,71]. Connectivity metrics alone, based on MSPA, Network Analysis and functional indices (e.g., IIC) have been instead commonly applied for setting habitat and corridor conservation priorities [46,72,73]. As an immediate advantage, the merge between environmental stratification and connectivity metrics mitigated the effect of patch area on node importance assessment [73,74], so that it was possible to include other nodes than the largest ones among the conservation priorities, bringing out their role as potential stepping stones. In spite of this benefit, implementation showed some limitations concerning EUNs affected by striking fragmentation. This is the case for the VPL unit in the southeastern sector of the study area, where only a few and low-priority conservation nodes could be identified and only 2 of the 20 restoration links were designed. Such a result suggests that conservation and restoration priorities should be framed in the first place on the difference between actual and potential cover of natural ecosystems, and only secondarily on the spatial pattern of remnants, as already proposed for the assessment of ecosystem conservation status at the national and regional level [42]. As regards the adopted connectivity indicators and with respect to consolidated practice [45,75], an approach not just based on node importance and link removal function allowed the contribution made by further elements of the landscape mosaic to be enhanced. Namely, branches and islets were explicitly included among the priorities so that their potential role as either stepping stones, discontinuous corridors, or habitat providers themselves [76–78] has been explicitly recognised while planning for conservation measures.

Some limitations could arise from the subjective choice of priority scores for each of the indicators, which however is often accepted as necessary in GI planning and could be eventually mitigated by including stakeholders and other disciplinary competences into the process [79]. The proposed restoration options are affected by a certain subjectivity as well. Nevertheless, these options comply with the evidence that new wooded links bring more benefits than converted ones and foster rodent dispersal, including that of *Apodemus* specimens [14,80–82]. Above all, however, the criteria for limiting as much as possible the consumption of productive space may facilitate, more than a more automatic but unfiltered least-cost path approach [83], the avoidance of potential conflicts with agricultural practices and the long-term persistence of planned interventions [49].

## 5. Conclusions

A set of criteria is presented for estimating and improving ecological connectivity at fine scales and that may be critical for planning effective GI in agricultural landscapes. The evidence provided by the implementation of these criteria in a peri-urban metropolitan sector emphasizes the usefulness of the ecological classification of land according to both the physical features of the environment and the biotic vegetation potential, and also provides a rationale for investing in detailed spatial representation and assessment of ecosystems. Notwithstanding the recognised limits, posed by the investigational character of the work but that can be quite easily disentangled in the case of a concrete GI deployment, it is hoped that the suggested approach will give useful hints for the requalification of transitional urban–rural areas and for the achievement of related sustainability goals, especially those prompted by the Green Infrastructure and Farm to Fork Strategies in Europe and by the Urban Green Strategy and the "Climate Decree" (national law decree n. 111/2019) in Italy.

**Supplementary Materials:** The following are available online at www.mdpi.com/article/10.3390/land10080807/s1, Table S1: adopted definitions for the concepts of ecosystem naturalness and ecological connectivity

**Author Contributions:** Conceptualization, G.C. and S.V.; Methodology, S.V. and G.C.; Investigation, S.V. and G.C.; Resources, G.C.; Data Curation, S.V., G.C. and L.Z.; Writing–Original Draft

Preparation, G.C. and S.V.; Writing–Review & Editing, S.V., G.C. and L.Z. All authors have read and agreed to the published version of the manuscript.

**Funding:** This research received no external funding.

**Data Availability Statement:** Some publicly available datasets were analyzed in this study, as reported in the reference section (references number 40 and 41). The new data were created in this study are available on request from the corresponding author.

**Acknowledgements**: The authors would like to thank the research unit on "Socio-ecological Systems, Landscape and Local Development "of the Complutense University of Madrid (UCM) for the support in applying the Kendall Tau-b statistic and for expert advice on linear element restoration criteria.

**Conflicts of Interest:** The authors declare no conflict of interest.

**Abbreviations**

AV = Alluvial Valleys; EUN = Environmental Unit; GI = Green Infrastructure; IIC= Integral Index of Connectivity; MSPA = Morphological Spatial Pattern Analysis; NC = Number of Components; PNV =Potential Natural Vegetation; SCS = Sandy-Clayey Slopes; VPL = Volcanic Plat-eaux.

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
