# Peer review of "Ecological Connectivity in Agricultural Green Infrastructure: Suggested Criteria for Fine Scale Assessment and Planning"

_land, doi:10.3390/land10080807_

Round 1

Reviewer 1 Report

Dear authors,

from my perspective a well written paper.

What I recommend to add is an introducing table about main definitions such as for:

  • Ecosystem integrity
  • Funcitonality
  • Landscape connectivity
  • Fragmentation

As explanation, what are in your opinion the lower and upper limits for this.

Author Response

We would like to thank the Reviewer for the useful suggestion.

In order to make more clear the basic concepts adopted in the research, we changed the introducing sentence in the Abstract and added a table of definitions as supplementary material (recalled in the Materials and Methods section at lines 119-120).

Specifically, the terms “ecosystem integrity” and “ecosystem functionality” have been removed from the abstract. Actually, they are respectively related to the concepts of “ecosystem naturalness” and “capacity to provide ecosystem services” we referred to, but they were not literally mentioned in the main text.

On the other hand, the table now provided in the supplementary material allows to clarify i) which definitions have been embraced with respect to ecosystem naturalness and ecological (structural and functional) connectivity , ii) what are the related concepts adopted for the presented research (i.e. vegetation condition, landscape conservation status, and landscape fragmentation) and iii) which are the scientific works we referred to as for these concepts.

As for the lower and upper limits of ecosystem naturalness we therefore implicitly referred to the scales reported in these reference works. Nevertheless, we better specified (in the supplementary table, but also in the main text ) that the potential natural vegetation at the occurring site represents the baseline against which the naturalness of actual vegetation can be assessed. In other words, the more a real vegetation community compositional and structural features are similar to those of an undisturbed one, the higher is its naturalness degree.

On the contrary, no lower and upper limits for ecological connectivity were reported because this landscape property may vary a lot with land contexts and observation scales and no reference values can be fixed a priori. For this reason, judgments on ecological connectivity are usually performed by comparison in time, that is by means of direct multi-temporal observations or by change scenarios (the latter option is that adopted in our research, as reported  at lines 248-251 of the revised manuscript).

Reviewer 2 Report

This is a very interesting article.The authors discuss the ecological connectivity of green infrastructure in agriculture, which has important implications for promoting biodiversity conservation and maintaining ecosystem integrity and functionality.The manuscript was well organized and able to appeal to a wide audience.I recommend it to be published after minor revision, and the suggestions are as follows:

(1) The reference space is too large, please modify it according to the standard format.

Author Response

We would like to thank the reviewer for the positive comment.

The reference list has been modified according to the standard format.

Reviewer 3 Report

The paper presents a methodology for improving ecological connectivity in peri-urban green infrastructure planning using a multi-step procedure. This is a very relevant topic. The paper is well written and the methodology is clear, but there is one major shortcoming. The authors state that they carried out an assessment of landscape naturalness, but they give no definition of ‘naturalness’, and no justification for classifying naturalness as either ‘high’ or ‘low’.  The objective of assessing naturalness is to identify landscape character and quality. Various approaches have been proposed and different vegetation classification methods have been applied to measure naturalness of landscapes. What method did the authors use?

Author Response

We would like to thank the Reviewer for the useful suggestion.

In keeping with the observation made also by Reviewer 1, we added a table (as supplementary material, recalled in the Materials and Methods section at lines 119-120) for a more clear definition of the basic concepts underlying the research, including that of ecosystem naturalness and the reference frameworks for its assessment.

Moreover, the sentence describing the criteria adopted for the assignment of a different naturalness degree to the woody components occurring in the observed landscape has been made more explicit (lines 154-159 of the main text): “Specifically, naturalness have been assessed accounting for the physiognomic and structural features of the mapped woody elements with respect to those of the PNV [42] (step 1c, Figure 2; Table S1 of Supplementary Material): areal and linear elements dominated by non-native species and/or with a regular structure due to plantation activities were considered less natural than those dominated by the native species typical of the PNV and showing a spontaneous cover pattern.”

We hope that these corrections made more clear that naturalness was not assigned to the overall landscape but just to its individual ecosystem components, and exclusively to those dominated by woody species (linear and areal shrub and forest cover types).

A comprehensive assessment of the landscape conservation status could be actually performed if a naturalness degree had been assigned to all the occurring land cover types (including for example artificial surfaces and agricultural areas). Nevertheless, we argued that a ‘connectivity-based’ rather than a ‘naturalness-based’ landscape condition assessment was more useful for achieving the research objectives. For this reason, we limited the assignment of a naturalness degree just to the landscape components assumed as suitable habitats for oak persistence and dispersal (as stated at lines 151-152 of the revised manuscript).